# Reducing *MSH4* copy number prevents meiotic crossovers between non-homologous chromosomes in *Brassica napus*

Adrián Gonzalo[1,3], Marie-Odile Lucas[2], Catherine Charpentier[1], Greta Sandmann[1], Andrew Lloyd[1,4] & Eric Jenczewski [1]

In allopolyploids, correct chromosome segregation requires suppression of non-homologous crossovers while levels of homologous crossovers are ensured. To date, no mechanism able to specifically inhibit non-homologous crossovers has been described in allopolyploids other than in bread wheat. Here, we show that reducing the number of functional copies of *MSH4*, an essential gene for the main crossover pathway, prevents non-homologous crossovers in allotetraploid *Brassica napus*. We show that non-homologous crossovers originate almost exclusively from the *MSH4*-dependent recombination pathway and that their numbers decrease when *MSH4* returns to single copy in *B. napus*; by contrast, homologous crossovers remain unaffected by *MSH4* duplicate loss. We also demonstrate that *MSH4* systematically returns to single copy following numerous independent polyploidy events, a pattern that is probably not by chance. These results suggest that stabilization of allopolyploid meiosis can be enhanced by loss of a key meiotic recombination gene.

[1] Institut Jean-Pierre Bourgin, INRA, AgroParisTech, CNRS, Université Paris-Saclay, 78000 Versailles, France. [2] INRA UMR1349 Institut de Génétique, Environnement et Protection des Plantes, Le Rheu 35653, France. [3] Present address: Department of Cell and Developmental Biology, John Innes Centre, Norwich NR4 7UH, UK. [4] Present address: Institute of Biological, Environmental, and Rural Sciences, Aberystwyth University, Aberystwyth SY23 3EB, UK. Correspondence and requests for materials should be addressed to E.J. (email: eric.jenczewski@inra.fr)

Polyploidy (Whole-Genome Duplication, WGD) has played a pervasive role in the evolution of many living organisms[1]. Many eukaryotes descend from polyploid ancestors that experienced massive duplicate gene loss over millions of years and ultimately reverted to a diploid state[1]. The long-term survival of newly formed polyploids depends, however, on their ability to withstand the extensive genomic instability that accompanies the onset of polyploid formation. Most notably, nascent allopolyploids, which result from interspecific hybridization followed by WGDs, require meiotic adaptation to safeguard genome stability and fertility; otherwise, crossover formation between the chromosomes inherited from the allopolyploid's progenitors (i.e. the homoeologues) will disrupt chromosome segregation and result in the production of aneuploid gametes[2,3]. Stabilization of allopolyploid meiosis thus requires that inter-homoeologue crossover formation is prevented while levels of homologous crossovers are ensured.

How meiosis has adapted to cope with allopolyploidy has only been deciphered in allohexaploid wheat, where a duplication of the *ZIP4* gene within the *Ph1* locus prevents maturation of crossovers between non-homologous chromosomes[4–7]. ZIP4 is an essential factor for the main crossover pathway[8,9] (called the class I or ZMM pathway) that also includes a set of other critical proteins (e.g., MER3, MSH4, MSH5, SHOC1, HEI10, PTD) in plants. While there are numerous examples of allopolyploids that have adapted meiosis to prevent homoeologous crossovers, no other gene has been identified that influences the rate of homoeologous recombination in allopolyploids.

In a previous paper[10], we proposed that meiotic adaptation in established polyploids could involve 'fine-tuning' the progression or the effectiveness of meiotic recombination. One potential pathway to "fine-tune" meiosis in newly-formed polyploids is through gene loss. This process has recently gained attention as a driver of functional innovation[11,12] and massively affects the evolution of recent polyploid genomes in a process referred to as *fractionation*[13]. Genes involved in meiotic recombination (and DNA repair in general[14,15]) seem especially susceptible to fractionation as they tend to return to a single copy more rapidly than genome average following subsequent and/or independent duplications[10]. In this paper, we ask whether the loss of some meiotic recombination gene duplicates can contribute to improved meiosis in these organisms.

To address this question, we first re-evaluate the patterns of duplicate gene loss and retention for different genes encoding proteins from the class I crossover pathway. We then describe how copy number reduction of *MSH4* (the ZMM gene that shows the most rapid fractionation) affects crossover formation in an allopolyploid (*Brassica napus*) to potentially enhance meiosis. *Brassica napus* (AACC, 2n = 38) is a recent allotetraploid crop bred for high seed yield, that originated from interspecific hybridizations between the ancestors of *B. rapa* (AA, 2n = 20) and *B. oleracea* (CC, 2n = 18) ~7500 years ago[16]. It is also, and most importantly, one of only two allopolyploid species (along with bread wheat[5–7]) for which meiosis has been thoroughly analyzed[17]. We focused on crossover formation between both homologous and homoeologous chromosomes; the latter, which are rare events in euploids (AACC), are readily quantifiable in allohaploid plants obtained by microspore culture[17]. These plants only contain one unique copy of each chromosome (AC, n = 19) so that (almost) all crossovers observed in allohaploids involve homoeologues.

Here, we show that regular meiosis is achieved in *B. napus* even when the number of functional *MSH4* copies is reduced to a minimum. By contrast, we demonstrate that the number of crossovers formed between homoeologues responds to *MSH4* dosage. This indicates that loss of one *MSH4* copy has no negative impacts and is potentially beneficial for *B. napus* meiosis, which improves when crossovers between homoeologues are suppressed.

## Results

**Convergent loss of duplicated recombination genes post-WGDs.** Building on previous work[10], we first measured the extent to which gene duplicates encoding plant ZMM proteins were lost or retained following >40 independent plant WGDs ranging in age from few thousand to more than 150 million years.

Four *ZMM*s show a conspicuous pattern of rapid duplicate gene loss following WGDs (Fig. 1; Supplementary Figs. 1–3) with a single gene encoding MER3, MSH4, MSH5 and ZIP4 proteins in almost all angiosperms other than the most recent polyploids (i.e., those that formed <10,000 years ago). In paleopolyploids that have returned to a diploid state post-WGD, the presence of duplicates for these four *ZMM* remains an exception usually limited to those with the next most recent WGDs: e.g., *MSH5* in *Glycine max* (5–9 million years old WGD[18]); *MER3* and *ZIP4* in *Linum usitatissimum* (5–13 million years old WGD[19]) (Fig. 1; Supplementary Figs. 1–3). By contrast, a slightly higher number of duplicates is observed among the genes encoding SHOC1, PTD and, most importantly, HEI10 (Fig. 1; Supplementary Figs. 4–6).

The most striking example of precipitous *ZMM* duplicate loss is *MSH4*, which shows only one intact copy in all diploid species (Fig. 1). To determine whether rapid *MSH4* duplicate loss post-WGD is specific to angiosperms, we expanded our analysis to include a taxonomically broad set of 95 fungi and 39 animal species that encompass 12 additional WGDs ranging in age from few thousand to almost 500 million years (Supplementary Figs. 7–8). Again, we identified only one intact copy of *MSH4* in all species except the very recent polyploids (*Zygosaccharomyces rouxii*) and *Diplocarpon rosae* and *Hortaea werneckii*, two fungi that have retained 80 and 95% of gene duplicates post-WGDs, respectively[20,21] (Supplementary Fig. 8). Remnants of a second copy of *MSH4* carrying inactivating changes were detected in only three plant (*L. usitatissimum*, *G. max* and *Populus trichocarpa*; Fig. 1c) and two animal (*Salmo salar* and *Oncorhynchus mykiss*; Supplementary Fig. 7) species. The inactivating changes found (important sequence gaps, mutations and truncations in exonic sequences) in these *MSH4* pseudogenes, are likely signatures of ongoing fractionation.

**B. napus carries two differentially expressed copies of MSH4.** The results above demonstrate that *MSH4* is a textbook case of the rapid loss of meiotic recombination duplicates following WGDs; they also indicate that, although very rapid on an evolutionary timescale (i.e., a few million years), *MSH4*-duplicate loss is not immediate. As shown in Fig. 1, *B. napus* contains two full-length *MSH4* homologues, hereafter referred as to *BnaA.MSH4* (marked as *Brassica napus* homoeo2 on Fig. 1) and *BnaC.MSH4* (marked as *Brassica napus* homoeo1 on Fig. 1). These two copies correspond to homoeologues *sensu*[22]. A partial (4 exons instead of 24) copy of *MSH4* was also found tandemly arrayed with *BnaA.MSH4* (Fig. 1; Supplementary Table 1). This additional copy is not transcribed during meiosis (based on *B. napus* meiotic RNAseq[23]) and was therefore not considered for further analyses.

We observed that *BnaA.MSH4* and *BnaC.MSH4* are both transcribed during male meiosis in the three varieties analyzed, with *BnaC.MSH4* contributing most to *MSH4* expression in *B. napus*; in all varieties, the balance between A/C contribution was in the range of 1/6 to 1/3 (Supplementary Table 2). This feature is unique to *MSH4* (compared to *MSH5*, *MER3* and *ZIP4*) and offers more opportunities to assess the effect of varied *ZMM*

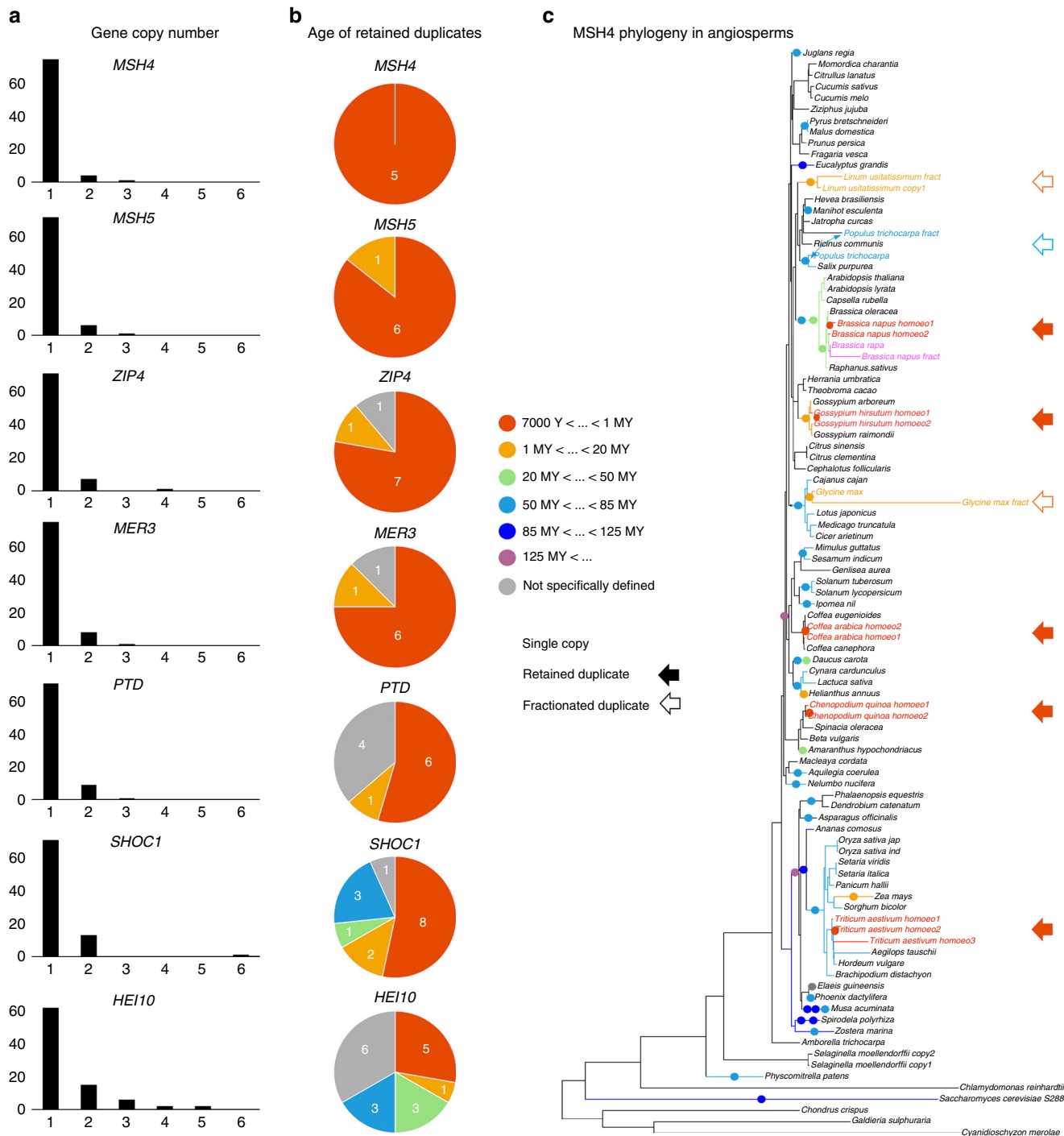

**Fig. 1** *ZMM* gene copy numbers in angiosperms. **a** Bar charts indicating the number of angiosperms showing one to six complete copies of the genes encoding each ZMM protein. Partial/fractionated copies are excluded. The different charts are based on a number of species ranging from 79 (MSH5) to 87 (HEI10). **b** Sector chart showing the age of the retained duplicates, as estimated by the age of the WGD they originate from. Only complete copies are considered here. Duplicates generated by small scale duplications (SSD) and/or that could not be associated with a specific WGD event are indicated in grey (i.e., age not specifically defined). **c** Maximum likelihood trees based on MSH4 amino acid sequences. For the sake of clarity, species names are indicated instead of gene names. Coloured disks superimposed along the branches of the tree give the age range for past WGDs (i.e., dark orange: 7000 years to 1 MY; light orange: 1 to 20 MY; light green: 20 to 50 MY; light blue: 50 to 85 MY; dark blue: 85 to 125 MY; violet: more than 125 MY). Full-length duplicates/triplicates (marked by solid arrows) and duplicates with one fractionated copies (marked by open arrows) are written with the colour that corresponds to the age of the WGD. Duplicates originating from SSD are written in magenta. Regarding the most recent WGD (i.e., dark orange: 7000 years to 1 MY), only allopolyploids are represented because there is no way to account for the presence of *MSH4* "duplicates" in extant autopolyploids, which truly correspond to (multiple) real alleles. The connectors superimposed over the phylogeny associates the full-length and fractionated copies of *MSH4* in *Populus trichocarpa*

dosage on crossover formation (see below). We, therefore, focused on this gene for further analyses.

The predicted amino acid sequences of *BnaA.MSH4* and *BnaC.MSH4*, consolidated using *B. napus* meiotic RNAseq data[23], both contained canonical MutS domains including the highly conserved C-terminal MutSac domain, which is essential for ATPase and DNA-binding activities[24] (Supplementary Fig. 9). We identified and selected using TILLING (Supplementary Fig. 10) one mutant allele in *BnaA.MSH4* (hereinafter referred as to *bnaA.msh4-1*, symbolized by $A^1$) and two mutant alleles in *BnaC.MSH4* (hereinafter referred as to *bnaC.msh4-1*, symbolized by $C^1$, and *bnaC.msh4-2*, symbolized by $C^2$, respectively). These alleles were the only ones for which the conserved MutSac domain was disrupted by early stop codons introduced by a point-nonsense mutation (*bnaC.msh4-1*) or because of splice site mutations (*bnaA.msh4-1* and *bnaC.msh4-2*) (Supplementary Fig. 11).

**Homologous crossovers largely depend on MSH4 in *B. napus*.** To ensure that the identified mutations did indeed compromise MSH4 function, we first assessed whether plants carrying only *msh4* mutant alleles ($A^1A^1C^1C^1$ and $A^1A^1C^2C^2$: Supplementary Fig. 12) showed a severe reduction in crossover leading to the occurrence of numerous univalents (i.e. chromosome that failed to form crossovers) at metaphase I[25,26].

We first analyzed the progression of male meiosis in WT *B. napus* cv. *Tanto*, the accession in which *msh4* mutations were identified. Our cytological survey showed that meiosis was very regular in cv. *Tanto*, like in other *B. napus* accessions[17]. During prophase I, meiotic chromosomes condensed, recombined and underwent synapsis, the close association of two homologous chromosomes via the Synaptonemal Complex (SC), which was complete at pachytene. From diakinesis to metaphase I, 19 discrete bivalents were identifiable in all cells (Fig. 2a). They consisted of pairs of homologous chromosomes bound together by chiasmata, the cytological manifestation of meiotic crossovers. We estimated that 57% of the bivalents in WT were rings with both arms bound by chiasmata while the remaining 43% were rods with only one arm bound by chiasmata. Assuming that rod and ring bivalents had only one and two crossovers, respectively, we estimated a mean of 30.0 chiasmata per cell in $A^+A^+C^+C^+$ ($n$ = 28; Fig. 2, Table 1). This could be an underestimate, however, given that it is not possible to distinguish cytologically single from multiple crossovers clustered on a single arm. The second meiotic division then took place and produced balanced tetrads of four microspores (Late telophase in Fig. 2a).

In the $A^1A^1C^1C^1$ and $A^1A^1C^2C^2$ double mutants, the early stages of prophase I were similar to those of WT *B. napus* cv. *Tanto* (Fig. 2a). We confirmed that synapsis was complete in these plants, as demonstrated by immunolocalization of ZYP1 protein, a major component of the central element of the SC[27] (Supplementary Fig. 13). Meiotic defects became obvious at the end of meiotic prophase when the bivalent formation was strongly compromised in both $A^1A^1C^1C^1$ and $A^1A^1C^2C^2$ (see metaphase I in Fig. 2a, b and Supplementary Fig. 14). We observed a mean number of 20.4 univalents (53% of chromosomes) that coexisted with 8.8 bivalents in $A^1A^1C^1C^1$ (Table 1). Contrary to WT, the majority of bivalents were rods (84%). At metaphase I, the reduction in chiasmata frequency was thus very evident, dropping down to an average of 9.9 chiasmata per cell (Table 1). The univalents then segregated randomly, resulting in unbalanced tetrads (See late telophase II in Fig. 2a).

To test whether the *msh4* mutant alleles completely suppress MSH4 activity, we immuno-localized HEI10 and MLH1, two proteins that specifically mark the sites of MSH4-dependent (i.e., class I) crossovers at diakinesis[28]. We consistently counted 29 foci

for the HEI10 and MLH1 proteins (Table 1; Figs. 2, 3) in the $A^+A^+C^+C^+$ plants that only carries WT *MSH4* alleles but is sibling to $A^1A^1C^1C^1$. By contrast, in the $A^1A^1C^1C^1$ and $A^1A^1C^2C^2$ double mutants, the residual chiasmata observed were not marked by HEI10 or MLH1 foci (Table 1; Fig. 2b and Supplementary Fig. 14), suggesting the absence of class I crossovers in these plants. This indicated that $A^1A^1C^1C^1$ and $A^1A^1C^2C^2$ were null *msh4* mutants and that *bnaA.msh4-1*, *bnaC.msh4-1* and *bnaC.msh4-2* alleles encoded nonfunctional MSH4 proteins.

**Total crossover numbers are unaffected by *MSH4* copy number.** To examine the functional consequences of loss of *MSH4* copies, we characterized the meiotic behaviour of plants carrying different combinations of WT and mutant alleles (Supplementary Fig. 12). We verified beforehand that loss of one copy is not compensated by a transcriptional upregulation of the remaining *MSH4* WT alleles, a rare but not unheard phenomenon[29–31]. The relative contribution of *BnaA.MSH4* and *BnaC.MSH4* along with their summed expression suggested no obvious decay or transcriptional compensation between functional and mutant *msh4* alleles (Supplementary Fig. 15).

We then moved on to the cytological survey of the exact same plants. We observed that functional loss of *BnaA.MSH4*, the least expressed *MSH4* copy, or *BnaC.MSH4*, the most expressed *MSH4* copy, all resulted in a WT-like meiosis (Fig. 3; Supplementary Figs. 16 and 17). The $A^1A^1C^+C^+$, $A^+A^+C^1C^1$ and $A^+A^+C^2C^2$ plants all showed 19 bivalents at metaphase I (Supplementary Fig. 16) and the same number of HEI10 foci as the WT (Fig. 3; Supplementary Fig. 17). These results indicated that, irrespective of their unequal transcriptional contribution, *BnaA.MSH4* and *BnaC.MSH4* are both equally functional and able to complement one another. We then explored the extreme situation where *BnaC.MSH4* was completely depleted while only one allele of the *BnaA.MSH4* was functional. This plant, i.e., $A^+A^1C^1C^1$, carried the minimum functional *MSH4* dosage possible, with only one functional allele of the least expressed copy (e.g., >90% reduction in functional transcript). Despite this minimal composition, $A^+A^1C^1C^1$ showed the exclusive bivalent configuration and about 31 HEI10 foci per meiocyte as in WT (Fig. 3; Supplementary Figs. 16f and 17f). All these results indicated that normal class I crossover formation is not sensitive to *MSH4* duplicate loss, providing that (at least) one functional copy of *MSH4* is present in the plant.

**Homoeologous crossovers are affected by *MSH4* copy number.** We then focused on crossover formation between homoeologous chromosomes in allohaploid plants produced from the two F1 hybrids ($A^+A^1C^+C^1$ and $A^+A^1C^+C^2$; Supplementary Fig. 12). As in euploids, we first verified that there was no transcriptional compensation between functional and mutant *MSH4* alleles in allohaploids showing varied number and assortments of WT ($A^+$ or $C^+$) and mutant *msh4* alleles ($A^1$, $C^1$ or $C^2$) (Supplementary Fig. 15).

We then estimated that WT allohaploids ($A^+C^+$) showed on average 6.3 chiasmata distributed over a mean number of 1 ring and 4.4 rod bivalents (Fig. 4 and Table 1). Interestingly, we observed a slight but significant decrease in chiasmata frequency (compared to WT allohaploids $t$ > 4.3, $p$-value < 0.0001, according to $t$-test) when the least expressed copy was depleted; the mean number of chiasmata in $A^1C^+$ plants dropped down to 5.1 crossovers (Fig. 4; Table 1). A stronger and significant reduction in chiasmata was observed when the most expressed copy was depleted, down to 2.2 chiasmata per cell in $A^+C^1$ (compared to $A^1C^+$; $t$ > 8.5, $p$-value < 0.0001, according to $t$-test) and 2.7 chiasmata per cell in $A^+C^2$ (compared to $A^1C^+$; $t$ > 2.4,

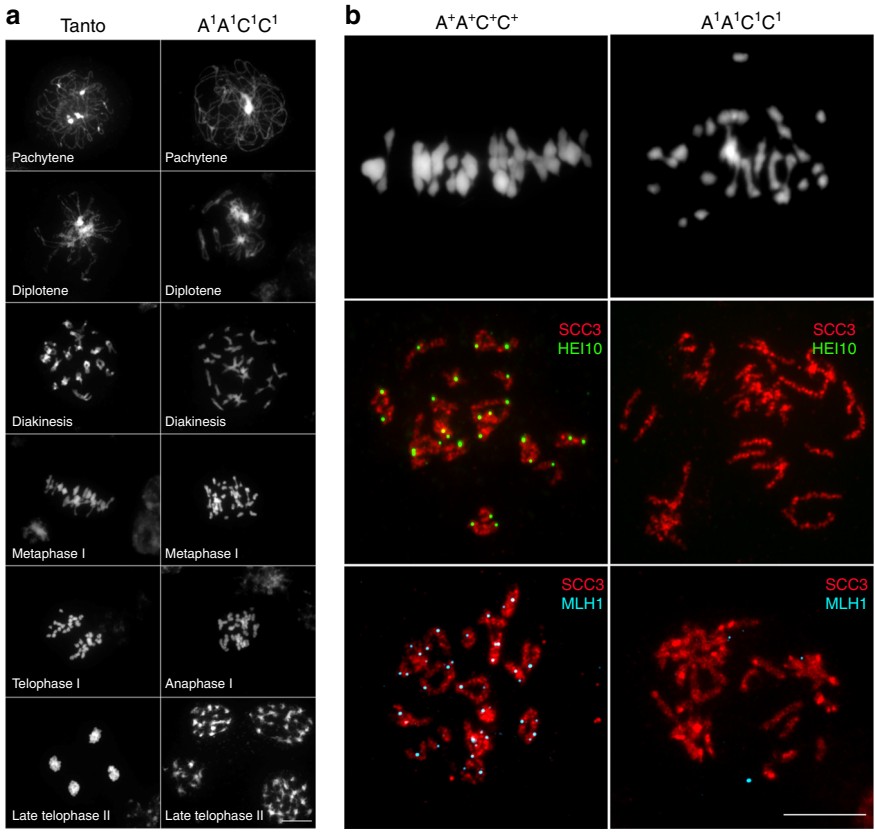

**Fig. 2** Meiotic progression and crossover formation in presence and absence of functional MSH4. **a** Comparative DAPI staining of wild type (*B. napus* cv *Tanto*) and $A^1A^1C^1C^1$ *msh4* double mutant. Different meiotic stages are illustrated: pachytene, diplotene, diakinesis metaphase I, Anaphase/telophase I, Late anaphase II, scale bar 10 μm. **b** Chiasmata and immunolabeled class I COs in WT and $A^1A^1C^1C^1$ *msh4* double mutant. The upper pictures show DAPI spreads of metaphase I. The middle pictures show dual immunolocalization of SCC3 and HEI10 at diakinesis stage. The lower pictures show dual immunolocalization of SCC3 and MLH1. Scale bar 10 μm

**Table 1 Crossover reduction in euploid and allohaploid *B. napus* msh4 mutants**

| Genotype | Number of bivalents | Number of chiasmata | Number of MLH1 foci | Number of HEI10 foci |
|---|---|---|---|---|
| $A^+A^+C^+C^+$ | 19 ± 0.0 ($n=55$) | 30.0 ± 2.0 ($n=28$) | 29.8 ± 4.9 ($n=8+9+21$) | 29.4 ± 2.6 ($n=47$) |
| $A^1A^1C^1C^1$ | 8.8 ± 2.5 ($n=62+27$) | 9.9 ± 2.4 ($n=62+27$) | 0.5 ± 0.5 ($n=19$) | 0.6 ± 1.7 ($n=39$) |
| $A^1A^1C^2C^2$ | nd | nd | 1.6 ± 1.4 ($n=15$) | 0.8 ± 1.1 ($n=36$) |
| $A^+C^+$ | 5.5 ± 1.5 ($n=51+38+37$) | 6.3 ± 1.9 ($n=51+38+37$) | nd | nd |
| $A^1C^+$ | 4.8 ± 1.5 ($n=59+15$) | 5.1 ± 1.7 ($n=59+15$) | nd | nd |
| $A^+C^1$ | 2.1 ± 1.3 ($n=33$) | 2.2 ± 1.4 ($n=33$) | nd | nd |
| $A^+C^2$ | 2.3 ± 0.6 ($n=3$) | 2.7 ± 0.5 ($n=3$) | nd | nd |
| $A^1C^1$ | 0.7 ± 1.1 ($n=51$) | 0.7 ± 1.1 ($n=51$) | nd | nd |
| $A^1C^2$ | 1.3 ± 1.6 ($n=18$) | 1.3 ± 1.6 ($n=18$) | nd | nd |

Summary of cytological estimation of CO frequencies as estimated by MLH1 or HEI10 foci and chiasmata frequency. Data are expressed as mean ± s.d. Sample size is given as a sum when several plants were analyzed for the same genotype
*nd* stands for not determined

*p*-value < 0.02, according to *t*-test) allohaploids, respectively (Fig. 4; Table 1). In these two plants, almost all bivalents were rods and only very rare ring bivalents were observed. Finally, when both *MSH4* copies were depleted, the number of residual chiasmata was further decreased, down to 0.7 in $A^1C^1$ (compared to $A^+C^1$; $t > 5.4$, *p*-value < 0.0001, according to *t*-test) and 1.3 in $A^1C^2$ (Fig. 4 and Table 1). Contrary to the euploids, immuno-localization of MLH1 and HEI10 proteins could not be used to confirm the decay of class I crossovers in all these plants. We have previously shown that in *B. napus* allohaploid meiosis the two

proteins do not always co-localize during meiosis in *B. napus* allohaploid[17], which makes Class I CO estimates technically impossible. This notwithstanding, our results indicated that inter-homoeologue class I crossover formation is sensitive to *MSH4* duplicate loss (Fig. 4).

## Discussion

In this paper, we have explored the possibility of dosage-driven effects of *ZMM* genes[32,33] in meiotic adaptation to allopolyploidy.

We first observed that acquisition of additional copies of *ZMM* genes by small scale duplications, as exemplified by wheat's *ZIP4*[4], are exceptional events (Fig. 1; Supplementary Figs. 1–6) and not a convergent phenomenon in plant polyploids. Rather, we found a consistent and rapid reduction of copy number for genes encoding MSH4, MSH5, MER3 and ZIP4 following independent WGDs, while *SHOC1* and *HEI10* showed higher duplicate retention rates (Fig. 1; Supplementary Figs. 1–6). HEI10 is the only plant ZMM protein known to affect homologous recombination in a dosage-dependent manner, with the number of ZMM-dependent crossovers between homologous chromosomes increasing and decreasing with increased and decreased

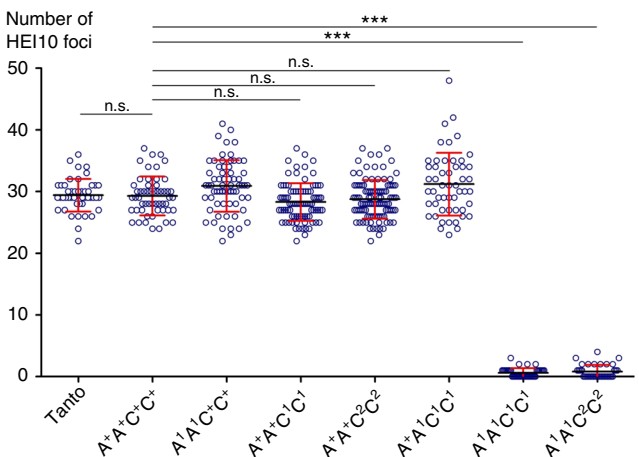

**Fig. 3** Robustness of ZMM-dependent crossovers to *MSH4* copy number reduction in *B. napus* euploids. Scatter plot of HEI10 (class I crossover) foci in the different genotypes, as determined by HEI10 immunolocalization. *** indicates highly significant variations (*p*-value < 0.0001) according to Kruskal–Wallis test. Error bars show standard deviation. Means (black bar) are also indicated

*HEI10* copy number, respectively[33]. The higher *HEI10* duplicate retention observed is therefore consistent with the most widely accepted theory that explains the fate of gene duplicates post-WGD and posits that selection to cope with gene dosage constraints contributes to prolonged duplicate retention[34,35]. The same theory predicts that "gene duplicates that are not under selection to be retained in duplicate post-WGD are lost at random[35]". *MSH4* genes, which returned to singletons in all species except the most recent polyploids (Fig. 1), may illustrate this second trend. According to some authors, however, such a consistent pattern of gene loss is unlikely to be the result of a purely random and neutral process, but rather suggests duplicates have negative fitness impacts and are actively selected against[14,15,35]. To gain concrete insights into the consequences of *MSH4* duplicate loss, we have interrogated meiosis in recent allotetraploid *B. napus* when *MSH4* duplicates return to single copy.

MSH4 accounts for the majority of meiotic crossovers in most organisms, including plants[25,26]. We first confirmed that MSH4 plays the same role in *B. napus* as in other plants: though not required for synapsis completion (Supplementary Fig. 13), MSH4 is essential to ensure normal crossover numbers between homologues (Table 1; Figs. 2 and 3) and, therefore, strictly required to ensure fertility. Most importantly, our results showed that normal levels of homologous crossovers are robust against *MSH4* gene duplicate loss even when only one single functional allele of the least expressed copy was present (in the $A^+A^1C^1C^1$ plant, Fig. 3; Supplementary Figs. 16f and 17f). Conversely, we showed that crossover formation between homoeologous chromosomes fluctuates in a dosage-sensitive manner. It is maximum when all *MSH4* copies are functional, decreases progressively with the number of copies and approximates zero when all *MSH4* copies are not functional (Fig. 4). Together, our results suggest that MSH4 is in excess during normal meiosis (euploids) but is rate-limiting for allohaploid meiosis. Why does MSH4 become limiting only when crossovers are formed between homoeologous chromosomes in allohaploids? A tentative explanation is that a greater cellular concentration of MSH4 is required to stabilize

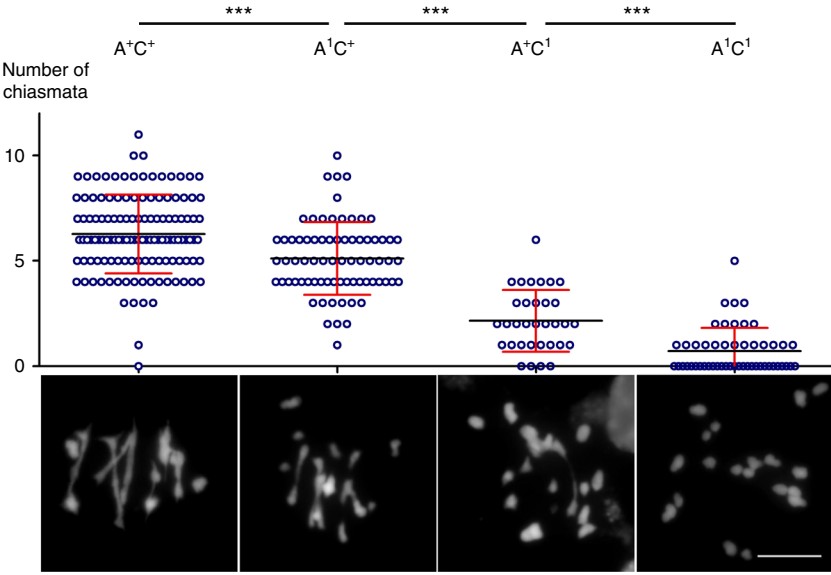

**Fig. 4** Decreased crossovers upon *MSH4* copy number reduction in *B. napus* allohaploids. Scatter plot showing the number of chiasmata estimated for allohaploids carrying varying doses of functional and *msh4* mutant alleles. Each genotype is illustrated below with a cytological image as example. N.s indicates non-significant variations whereas *** indicates highly significant variations (*p*-value < 0.0001), according to *t*-test. Error bars show standard deviation. Means (black bar) are also indicated. Scale bar 10 μm

inherently less stable/more transient early inter-homoeologue recombination intermediates that tend to abort before they mature into crossovers. Future work should test this mechanistic hypothesis.

This dispensability of *MSH4* duplicates (as long as one functional allele is present) shows that return of *MSH4* to single copy is unlikely to induce a selective cost in allopolyploids. Quite the contrary, our results suggest that *MSH4* duplicate loss could be beneficial. Providing that inter-homoeologue crossover formation in allohaploids is a good proxy for the situation in the euploids, where both homologues and homoeologues are competing, *MSH4* copy number reduction is likely to specifically reduce crossovers between homoeologous chromosomes and thus contribute to improved chromosome segregation. Taken alone, this observation, which is consistent with pervasive *MSH4* duplicate loss, would indicate that MSH4 is potentially a "major player" in the meiotic adaptation of allopolyploids. However, combined with the results from wheat (in which ZIP4 is the main actor), they could suggest that modulation of the entire the ZMM pathway, or at least part(s) of it, could contribute to meiotic stabilization in allopolyploids.

It is not clear whether MSH4 and ZIP4 act on the same step of the ZMM pathway, or even that their specific role is conserved between species. For example, although no ZMM crossovers can form when either MSH4 or ZIP4 are absent in *Arabidopsis*[8,25], rice meiosis seems to retain some residual ZMM-dependent events in *oszip4* mutant[9]. Likewise, the MSH4-MSH5 heterodimer is thought to act upstream of ZIP4 in rice[26,36], but it seems to be the opposite in yeast[37]. Finally, we do not know why inter-homoeologue crossover suppression results from *MSH4* duplicate loss in *B. napus* (this study) but *ZIP4* duplication in wheat[4]. These differences call for further investigation into the effect of the loss/gain of individual *ZMM* duplicates on crossover formation between homologous and homoeologous chromosomes in recent allopolyploids.

This notwithstanding, our results could lead to considering a simple and general route for meiotic adaptations in allopolyploids: i.e., through reduction in the efficiency of the ZMM pathway that would be substantial enough to prevent inter-homoeologue crossover formation but not to compromise the occurrence of (at least) one crossover per homologue (i.e., the obligate crossover). If that is the case, then there would be no need to acquire new meiotic function(s) to sort divergent chromosomes, as the basic machinery would be sufficient to achieve this outcome. In addition, beneficial variants of pre-existing meiotic of genes could occur with greater ease, or in greater proportion, as they would be mainly loss-of-function alleles: e.g., dominant-negative mutations that poison ZMM multiprotein complexes (e.g., MSH4-MSH5[38,39] or MSH5-ZIP4[37]) or nonsense / missense mutations that suppress or knock down the function of one gene. In any event, however, our result suggest that reduced cellular availability of MSH4 is unlikely to contribute autopolyploid meiosis stabilization, where the challenge is to prevent multivalent formation between chromosomes with little degree of differentiation. In this context, reducing the absolute number of crossover events, rather than improving their stringency (i.e., homologous vs homoeologous), plays a greater role[40–42]. The invariability of homologous crossover numbers with varying *MSH4* copy numbers (Fig. 3, Supplementary Figs. 16 and 17) is not conducive to this process. This difference justifies why genome scans in diploid and auto-tetraploid *Arabidopsis arenosa* populations revealed that differentiation (rather than duplicate loss) of structural components of meiotic chromosomes (rather essential factors for ZMM pathway) likely mediates autopolyploid stabilization[41].

To conclude, our results not only shed new light on the longstanding conundrum of meiotic adaptation in polyploids,
they also open fruitful avenues towards future applications. These arise from a better understanding of the particular mechanisms that govern recombination between differentiated chromosomes, such as stabilization of synthetic allopolyploids[43,44] or manipulating recombination-associated diversity in certain hybrid contexts[45].

## Methods

**Copy number assessment and phylogeny construction**. We first constructed an orthologous set of ZMM homologous proteins identified from BLASTP and TBLASTN searches against the Non-redundant protein sequences database of NCBI, the predicted protein sequences or current genomes assemblies available in EnsemblPlants (http://plants.ensembl.org/index.html), Plaza v4.0[46] (https://bioinformatics.psb.ugent.be/plaza/), Phytozome v12 (https://phytozome.jgi.doe.gov/pz/portal.html), MycoCosm (https://genome.jgi.doe.gov/programs/fungi/1000fungalgenomes.jsf) and/or some other website specifically dedicated to a group of related species (e.g. https://solgenomics.net/; https://sunflowergenome.org/). The sources of the amino acid sequences are provided in Supplementary Data 1.

Gene trees were reconstructed using the Maximum-likelihood (ML) method implemented in the phylogeny pipeline provided by Phylogeny.fr (http://www.phylogeny.fr/)[47]. Multiple alignments were carried out with MUSCLE (full mode). Alignment curation was implemented using GBLOCKS, allowing smaller final blocks and gap positions within the final blocks. Phylogenetic trees were generated using PhyML v3.0 after estimating the Gamma distribution parameters, the proportion of invariables sites and the transition/transversion ratio. The trees were customized and annotated using iTOL (https://itol.embl.de/).

The occurrence and age for the past WGDs were taken from refs. [1,18,19,48–52]. We used (i) collinearity information (i.e., location within conserved block) as inferred from the WGDotplot applet of PLAZA 4.0 and/or the SynMap applet of the CoGe platform (https://genomevolution.org/CoGe/SynMap.pl), (ii) phylogenetic information (using a subset of related proteins) and/or (iii) Ks (i.e., number of synonymous substitutions per synonymous site) estimation to determine whether a gene pair has arisen from a given WGDs[10]. If in doubt, the pair of duplicates was considered as "non-assigned".

**Plant material**. *Brassica napus* L. cv. *Tanto* is a double-low spring cultivar obtained at INRA Rennes (France). This is the accession used to develop the RAPTILL EMS population (described in ref. [53]), in which we looked for mutations in *BnaA.MSH4* and *BnaC.MSH4*.

Microspore culture was performed to isolate allohaploid plants following the protocol detailed in ref. [54]. Microspores were extracted from buds between 3.8 and 4.5 mm in length previously sterilized in bleach and suspended at a density of 100.000 microspores/mL in a modified Lichter's medium with 0.5 mg/L naphthaleneacetic acid, 0.05 mg N6-benzyladenine, and 13% sucrose and without potato extract. This suspension was incubated 24 h in the dark at 4 °C and, after centrifugation and replacement of the old medium by a fresh one, was incubated on plates in the dark at 30 °C for 14 days during which embryo development takes place. After this period, plates were moved to a slow gyratory shaker (60 rpm) and incubated in the dark at 28 °C for an additional week. After this incubation, embryo germination was induced in B5 liquid medium with 0.1 mg/L gibberellic acid and 1% sucrose incubated on a gyratory shaker (80 rpm) at 25 °C in light for 10 days. Next, the embryos were cultivated in long day greenhouse conditions (see below) on B5 solid medium (0.8% agar) with 0.1 mg/L gibberellic acid and 2% sucrose until root development allowed transference of the plants to soil. Ploidy level verification is recommended as few diploids might have been regenerated. All plants (euploids and allohaploids) were cultivated in standard long day greenhouse conditions (16-hour light/8-hour night photoperiod, at 22 °C day and 18 °C night).

**Identification of Brassica *MSH4* homologues**. *MSH4* homologues were identified by querying the *MSH4* coding sequence of *A. thaliana* (AY646927) against the reference assemblies of *B. rapa*[55] (http://brassicadb.org/brad/blastPage.php), *B. oleracea*[56] (http://plants.ensembl.org/Multi/Tools/Blast?db=core) and *B. napus*[16,57] (http://www.genoscope.cns.fr/blat-server/cgi-bin/colza/webBlat and http://appliedbioinformatics.com.au/gbrowseblast/cgi-bin/index.pl). Sequence alignments then confirmed that *BnaA.MSH4*/*BnaA08g08260D* and *BnaC.MSH4*/*BnaCnng35120D* correspond to the two *MSH4* genes isolated by previous screening a BAC library from *B. napus* cv. *Darmor-bzh*[10].

**TILLING**. We screened the RAPTILL EMS population following exactly the same protocol as described in ref. [53]. Briefly, we targeted a region of 1 Kb of the MutSac domain of *BnaA.MSH4* and *BnaC.MSH4*. We used copy-specific primers (T_MSH4AF1, T_MSH4AR1, T_MSH4CF1 and T_MSH4CR1; Supplementary Table 3) to amplify *BnaA.MSH4* and *BnaC.MSH4* separately. The resulting amplicons span the regions between the positions 4144–5126 and 4161–5148 in the genomic sequences of *BnaA.MSH4* and *BnaC.MSH4*, respectively. The screens for mutations then implemented the PMM (Plant Mutated on its Metabolites) method[58,59].

We produced two different F1 hybrids that combined *bnaA.msh4-1* (A[1]), the mutant allele selected for *BnaA.MSH4*, with either *bnaC.msh4-1* (C[1]) or *bnaC.msh4-2* (C[2]), the two mutant alleles selected for *BnaC.MSH4*. These F1s were self-fertilized to produce segregating F2 progenies among which we selected plants that contained varied number and assortments of Wild Type (A[+] or C[+]) and mutant *msh4* alleles (A[1], C[1] or C[2]) (Supplementary Fig. 6).

**Genotyping**. The mutant alleles were identified using cleaved amplified polymorphic sequences (CAPS) assay targeting the causative EMS-SNP. The list of primers and restriction enzymes in given in Supplementary Tables 3 and 4.

**Cytological analysis**. Staged anthers were isolated from fixed buds in ethanol:acetic acid 3:1 (v/v) and used to produce meiotic spreads as described in detail in ref. [60]. Briefly, fixed anthers were firstly rinsed in 10 mM citrate buffer pH 4.5, incubated by 3.5 h at 37 °C in a digestion mix (0.3% cellulase, 0.3% pectolyase Y23 and 0.3% cytohelicase in citrate buffer pH 4.5). Individual digested anthers were carefully rinsed in water and homogenized in a drop of water placed on a microscopy glass slide. After adding 20 μL of 60% acetic acid the resulting drop was stirred during 4 min on a hot plate at 45 °C to remove the cytoplasm of cells. Finally, the drop was fixed by pipetting etanol:acetic acid (3:1) and allowing the slide to dry before mounting the slide. Chiasma number was estimated on metaphase I spread chromosomes counterstained with 4′,6-diamidino-2-phenylindole (DAPI) based on bivalent shape. Open "rod" bivalents were considered to contain one single chiasma while closed ring bivalents were scored as two (one on each arm). For the study of allohaploids, all scorings were done blindly. Spreads were also used for immunolocalization. Raw counts of bivalent, univalent and chiasma numbers are provided in Supplementary data 2.

**Immunolocalization of meiotic proteins**. Immunodetection of ASY1 and ZYP1 followed the protocol described by ref. [8] that uses fresh anthers treated with 1% lipsol and fixed in 4% paraformaldehyde on a slide. These slides were washed in 0.1% Triton in PBS (PBS-T) and incubated with the corresponding antibodies diluted in 1% BSA at 4 °C for 24 h. Finally, slides were washed in PBS-T were done before mounting with DAPI plus Vectashield.

Immunodetection of MLH1, HEI10, REC8 and SCC3 proteins was performed following the protocol described in detail in ref. [61]. This protocol requires a microwave treatment of recently fixed spreads at 850 W in citrate buffer pH 6 that ends by transferring the slides immediately to PBS-T. After this treatment slides followed the antibody incubation and all the successive steps described for standard immunodetection with the difference of a longer antibody incubation (48–72 h). The anti-MLH1 and anti-HEI10 antibodies, both obtained in rabbit serum, were used at dilution 1:200. The antibodies raised against SCC3 and REC8, from guinea pig (GP) and rat serum, respectively, were diluted at 1:250. Anti-ASY1 and anti-ZYP1, obtained in guinea pig (GP) and rabbit, respectively, were both used at dilution 1:500. Secondary antibodies were Alexa488-anti-rabbit (green) and Alexa568-anti-GP and -anti-rat (red). Raw counts of HEI10 foci are provided in Supplementary Data 2.

**Fluorescent microscopy**. All images were obtained using a Zeiss AxioObserver microscope and were analyzed by means of Zeiss Zen software and were organized in panels with Adobe Photosoft CS3.

**RNA extraction and cDNA obtention**. RNA was extracted from buds selected by size using RNeasy Mini Kit (Qiagen). One extra step of clean-up and DNAse treatment was added to the extraction protocol. Resulting RNA was utilized for cDNA obtention using RevertAid First Strand cDNA Synthesis Kit (Thermo Fisher Scientific). The presence and absence of cDNA and residual genomic was assessed using the primers Q_UBC21R1, Q_UBC21L1, Q_MSH4F1 and Q_MSH4R1 (Supplementary Table 3). Euploid and allohaploid material was sampled simultaneously under the same conditions.

**Quantitative PCR**. Three biological replicates were analyzed for every euploid genotype, each replicate being located on a separate qPCR plate. For the allohaploids, a total of five plants were used. Three of them, each representing a distinct genotype, were distributed among three different plates. The last two plants, which represented the same fourth genotype, were distributed on two and one plate, respectively. Three technical replicates were used for all (euploid and allohaploid) plants, all located onto the same plate. We used the ubiquitin gene *UBC21* as a reference to normalize the expression of the target *MSH4*. The primers used in these experiments (Q_UBC21R1, Q_UBC21L1, Q_MSH4F1 and Q_MSH4R1) are given in Supplementary Table 3. Variation of *MSH4* expression level between genotypes was assessed using an ANOVA with two fixed effects (Genotype and Plate) and the difference in threshold cycle between *MSH4* and *UBC21* as the dependent variable. Pairwise comparisons of samples after ANOVA were done using post-ANOVA Tukey's test (alpha = 0.05).

**Pyrosequencing**. For pyrosequencing, cDNA and genomic DNA was used to test for amplification bias. The primers utilized were PS_MSH41F (5′-TGCTCAGA

TTGGCTGCTATGT-3′) as forward primer, PS_MSH4R (5′-TTCTTGTGAATA TGCGGTCAAC-3′) as reverse primer and PS_MSH4S (5′-CAACCACACGCAT AGT-3′) as sequencing primer. Pyrosequencing reaction was performed with PyroMark Q24 v2.0.6 (Qiagen).

**Reporting summary**. Further information on research design is available in the Nature Research Reporting Summary linked to this article.

## Data availability
The data that support the findings of this study are available from the corresponding author upon reasonable request. Raw data required to produce the figures and graphs shown in this work are provided in Supplementary Data 2.

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

## Acknowledgements

We would like to thank Karine Alix, Christine Mézard, Raphaël Mercier and Mathilde Grelon for critical reading and discussion of the manuscript. We also thank Fatiha Benyahya for her help and acknowledge AELRED for performing the TILLING experiment in *B. napus*. This work was funded through the ANR project ANR-14-CE19-0004 – CROC. The IJPB benefits from the support of the LabEx Saclay Plant Sciences-SPS (ANR-10-LABX-0040-SPS). A.G. was funded by the Marie-Curie "COMREC" network FP7 ITN-606956. G.S. is funded by the Marie-Curie "MEICOM" network H2020 ITN-2017-765212. A.L. was funded by the International Outgoing Fellowships PIOF-GA-2013-628128 POLYMEIO.

## Author contributions

Study design: A.G., A.L. and E.J.; Plant material production: A.G., M.O.L. and C.C.; Cytological analyses: A.G.; Transcriptome analyses: A.G. and A.L.; Phylogenetic and statistical analyses: E.J.; G.S., A.G., A.L. and E.J. contributed to writing the manuscript.

## Additional information

**Competing interests:** The authors declare no competing interests.

