## [Peer Review File · Nature Communications]

Reviewers' comments:

Reviewer #1 (Remarks to the Author):

The authors have quite possibly identified the genetic factor responsible for prevention of non-homologous chromosome pairing in Brassica allopolyploids, thus contributing to allopolyploid genome stability. These results are the first in any animal or plant polyploid system other than bread wheat, and the first in the dicots; the finding that copy number variation of a meiosis gene is responsible for genome stability matches the finding in wheat but with a different gene involved, and with loss rather than gain. This is really quite exciting for the polyploidy and evolutionary biology community and suggests fruitful avenues for research into polyploid genome evolution in other species. I have the following comments and suggestions for improvement.

1. In general I think the paper is quite technically, rather than generally, written. This is not really a problem but in order to reach and interest a wider audience I think it could be made really more explicit why these results are exciting and novel, particularly in the Introduction and Discussion. Quite often, mention is made of particular research questions or hypotheses without much context about why these are important or interesting.

2. I would prefer that the "story" of the paper is about meiotic adaptation in polyploids, and not about the genome-wide fate of duplicated genes and hypotheses related to this. Analysis of one meiosis gene (when we already know from previous work of the authors that meiosis genes tend to revert to single copy numbers faster than other genes) is unlikely to provide convincing support for hypotheses related to genome-wide gene loss (e.g. as mentioned at the start of the Discussion). There is also no guarantee that the same pressures operate on e.g. meiosis genes as opposed to e.g. flowering time genes or disease resistance genes; putatively some gene networks would be expected to have greater tolerance of copy number variation than others, depending on how strictly controlled the phenotype must be for plant survival and viability, and how likely variation is to provide an active substrate for environmental selection. I find generalisations in this direction rather unconvincing, and I'm not sure that there will prove to be a general rule for all gene families from a genomics perspective.

3. Why was only MSH4 investigated/reported on? What about other meiosis genes? The authors need to provide some more justification for why only MSH4 was investigated in the introduction, or if other genes were also considered (at least bioinformatically) then this information should be in the results or supplementary information. There must have been some rationale based on functionality etc., unless all the meiosis genes mentioned were investigated and then this one turned out to be the key candidate.

4. Please stick to the scientific conventions of what goes in the introduction, methods, results and discussion sections. Currently, quite a lot of the information in the results sections is really methods or introduction sentences. I understand this is difficult based on the format of having the methods section last, but I would prefer these results parts were condensed to just present the important results points here, rather than including too much description and justification of the experimental process involved. I'm guessing this was revised from a combined submission format. Regardless I think that most readers without the specific technical background will find it harder to understand results that are also interspersed with methods, since methods tend to be quite field-specific, and will be more interested in the results rather than the methods used to obtain them.

5. Can the authors comment in the Discussion on how important they predict the MSH4 gene to be in stabilizing meiosis in allopolyploid *B. napus*, and how generalizable these results might be to other allopolyploids? What about the relation of the results to the extra copy of ZIP1 at the Ph1 locus in wheat? I would like to hear more (i.e. expand this Discussion section) about whether any of these previous results are related in any way to the current findings related to the MSH4 gene – do the genes interact or fulfil similar functions? Can the authors predict what we would expect to

find in other polyploid species, or what genes they think are potentially the “major players” in polyploid stabilization?

6. In *Arabidopsis*, selective sweeps found several genes potentially responsible for restoration of genome stability via reduction of crossover number in tetraploids relative to diploids. Can the authors elucidate the differences between the genes and mechanisms involved between prevention of non-homologous crossovers and reduction in crossover frequency? Are both likely to operate in allopolyploids, or is the latter predicted to be autopolyploid specific?

Figures and Tables

7. Figure 2, panel 3 in the top row – does the cluster of chromosomes in the middle of the picture represent bivalents or univalents? Is a clearer image available, or is this a function of the mutant phenotype? The same image is in Supplemental Figure 7

8. Are parts A and B of Figure 3 associated with any statistical significance? Why is “Genome” there in part A? If the point is just to demonstrate that there are no differences between categories, I don’t think it’s necessary to include Parts A and B at all. Just Part C is fine if you want to show the double mutant has a major effect.

9. I also don’t really think it’s so critical to show parts A and B of Figure 4. For Figure 4 have you considered aligning parts C and D over each other, so that the number of crossovers is then matched by an example picture for this mutant type above it? That would make a great single combination figure if it can be done effectively.

10. Is it the case that not all of the mutants were screened for the same things? The figures don’t seem to match with respect to inclusion of the different mutant types/combinations.

11. There are no x-axis or y-axis labels on some figures.

12. Typo in Figure 1 – “sinlge”

13. Table 1 – based on the number of cells counted, it’s statistically invalid to include such a high degree of accuracy as is indicated by the number of decimal places. Even one decimal place is a bit high to be honest, but I’d suggest just going with maximum 2 significant figures.

14. The title for Table 1 is not descriptive enough for the information to stand on its own – needs species name, basic mutant description etc.

15. It took me a really long time to find and understand the key for Figure 1, this is really not so clear. I recommend that parts B – D of this figure move to the supplementary information, and that the key format for the indicator of whether the gene copies were duplicated or not is presented using just straight lines instead of curved, and in a table-like format next to the age of WGD events key, preferably in the top right corner. Why are there two > symbols instead of one in the WGD timing key?

16. Most of the text is talking about results in the allohaploids, so it seems like it would make sense to merge Table 1 with Supplementary Table 3 and include this in the main manuscript.

17. I really like Supplemental Figure 5, although I’d like it better if the wild-type and double mutant meiosis stages were lined up next to each other for ease of comparison, and clearly labelled on the figure as to which was which. Any chance of including this in the main text, maybe by merging with Fig. 2 and cutting one of the two mutant types out from the existing figure 2 to the supplementary? Which double mutant is being presented in Fig S5?

Reviewer #2 (Remarks to the Author):

This is an interesting paper that addresses the role of a conserved recombination factor, MSH4, in meiotic recombination in a polyploid plant (*Brassica napus*). The paper opens with a consideration of the documented phenomenon where meiotic genes are known to reduce to single copy more rapidly than other genes, following whole genome duplication (WGD). As a case in point they present analysis of MSH4, which conforms to this pattern. The authors present a nicely performed phylogenetic analysis on this point - but I think it would be interesting to expand this more generally to other ZMM factors. For example, is the same trend evident for MSH5, HEI10, SHOC1, PTD1, ZIP4 etc?

Building from this premise the authors then perform genetic analysis of MSH4 in *Brassica napus* with increasingly fewer functional copies of the gene - ie they compare all combinations of msh4 loss of function alleles. The assay for meiotic recombination they perform is cytogenetic - specifically they assess the number of univalent, rod and ring form bivalents at metaphase I, in addition they score MLH1 and HEI10 foci, which are markers of Class I interfering crossovers. The authors show that reducing to even a single functional copy of MSH4 is sufficient to show a wild type level of crossover, although it is hard to assess from these data whether the landscape along the chromosomes is the same. They also perform expression analysis to show that there is an absence of compensatory changes in gene expression when MSH4 dosage is reduced. This conclusion is to some extent expected as where tested msh4 is known to be a recessive mutation in other species.

The most interesting aspect of the paper involves the authors generating AC haploid plants where the chromosomes lack true homologs to pair and interact with. However, in these plants some pairing and recombination is evident between homeologous chromosomes. In this case the authors do see a much stronger sensitivity to MSH4, which reductions in MSH4 dosage causing much greater reductions in cytological crossover. Can the authors say anything about the likely locations of homeologous crossovers along the chromosomes from their data, and whether this varies with reducing MSH4 activity?

These findings imply that ZMM dosage, in this case via MSH4, has a strong effect on homeologous recombination. The authors quite rightly relate this to recent findings that the wheat Ph1 locus contains a duplication of the ZIP4 ZMM gene, and that this locus has a strong effect on homologous vs homeologous recombination. Therefore, the authors findings have important relevance for regulation of recombination in hybrid situations, which are common in plants. It may also be worth mentioning that HEI10 dosage is also known to regulate levels of homologous recombination in *Arabidopsis* (Ziolkowski 2017). Overall the manuscript and data are clearly presented and well written.

Minor point: The title and abstract do not mention the species that this study focuses on (*B.napus*) - please add this information!

Reviewers' comments:

Reviewer #1 (Remarks to the Author):

The authors have quite possibly identified the genetic factor responsible for prevention of non-homologous chromosome pairing in Brassica allopolyploids, thus contributing to allopolyploid genome stability. These results are the first in any animal or plant polyploid system other than bread wheat, and the first in the dicots; the finding that copy number variation of a meiosis gene is responsible for genome stability matches the finding in wheat but with a different gene involved, and with loss rather than gain. This is really quite exciting for the polyploidy and evolutionary biology community and suggests fruitful avenues for research into polyploid genome evolution in other species. I have the following comments and suggestions for improvement.

1. In general I think the paper is quite technically, rather than generally, written. This is not really a problem but in order to reach and interest a wider audience I think it could be made really more explicit why these results are exciting and novel, particularly in the Introduction and Discussion. Quite often, mention is made of particular research questions or hypotheses without much context about why these are important or interesting.

These are fair comments. We have now re-written a very large part of the paper, most notably the introduction and discussion. We hope all these changes now facilitate reading and present our data more generally.

It is also true that we could have done a better job in highlighting and promoting our most important results. We always tend to err on the side of caution without over-interpreting positive results. We thus thank reviewer 1 for his/her advices and stimulation to “step out of our comfort zone”. We hope that the manuscript now highlight why our results are “exciting and novel”.

2. I would prefer that the “story” of the paper is about meiotic adaptation in polyploids, and not about the genome-wide fate of duplicated genes and hypotheses related to this. Analysis of one meiosis gene (when we already know from previous work of the authors that meiosis genes tend to revert to single copy numbers faster than other genes) is unlikely to provide convincing support for hypotheses related to genome-wide gene loss (e.g. as mentioned at the start of the Discussion). There is also no guarantee that the same pressures operate on e.g. meiosis genes as opposed to e.g. flowering time genes or disease resistance genes; putatively some gene networks would be expected to have greater tolerance of copy number variation than others, depending on how strictly controlled the phenotype must be for plant survival and viability, and how likely variation is to provide an active substrate for environmental selection. I find generalisations in this direction rather unconvincing, and I’m not sure that there will prove to be a general rule for all gene families from a genomics perspective.

Good points. Actually, we have hesitated a lot about the best “story” to tell. We completely agree with reviewer 1’s concern about the scope of our results; our point was that there are a lot of theories about the fate of gene duplicates, but little empirical data to support them. Our objective was thus to add a useful (yet very specific) contribution, but we agree that it is hard to generalize from the study of a single gene; hence our very cautious tone, when we proposed a hypothesis rather than an established fact in the previous discussion.

We have now followed reviewer 1’s suggestion to focus on meiotic adaptation in polyploids and completely re-written the paper in that direction.

3. Why was only MSH4 investigated/reported on? What about other meiosis genes? The authors need to provide some more justification for why only MSH4 was investigated in the introduction, or if

other genes were also considered (at least bioinformatically) then this information should be in the results or supplementary information. There must have been some rationale based on functionality etc., unless all the meiosis genes mentioned were investigated and then this one turned out to be the key candidate.

Why was only MSH4 investigated/reported on? MSH4 was chosen a long time ago, far before the ZIP4/Ph1 story was published, because this is: (i) one of the ZMM genes that show the most rapid fractionation (stated in the introduction) and (ii) the only ZMM protein that is encoded by two differentially expressed copies in *Brassica napus*, with the C copy contributing most to MSH4 expression in this allopolyploid species. We reasoned that this could provide more opportunities to assess the effect of varied dosage, a gamble that has eventually paid off.

What about other meiosis genes? MSH4 is the first ZMM protein for which we (and anybody we are aware of) have isolated mutants in *Brassica napus* (and any other allopolyploid species). As requested by reviewer 1 and 2, we now provide an assessment of gene copy number for all plant ZMMs (in angiosperms). This shows that “four ZMMs show a conspicuous pattern of rapid duplicate gene loss following WGDs with a single gene encoding MER3, MSH4, MSH5 and ZIP4 proteins in almost all angiosperms other than the most recent polyploids (i.e. that formed <10 000 years ago)” and that “a slightly higher number of duplicates is observed among the genes encoding SHOC1, PTD and, most importantly, HEI10”. These two results are now stated as such at the beginning of the results and illustrated in Figure1, Supplemental Figure1 and Supplemental Figure 2 (detailed below; see answer to comment #15).

We agree with reviewer 1 that we must justify why we focused on MSH4 rather than another ZMM showing a rapid decay of duplicated copies post-WGDs. This justification is now placed in the introduction (see below) and at the beginning of the second section of the results. “We observed that *BnaA.MSH4* and *BnaC.MSH4* are both transcribed during male meiosis in the three varieties analyzed, with *BnaC.MSH4* contributing most to MSH4 expression in *B.napus* (Table S2). In all varieties, the balance between A/C contribution was in the range of 1/6 to 1/3 (Table S2). This feature is unique to *MSH4* (compared to *MSH5*, *MER3* and *ZIP4*) and offers more opportunities to assess the effect of varied ZMM dosage on crossover formation (see below). We thus focused on this protein.”

4. Please stick to the scientific conventions of what goes in the introduction, methods, results and discussion sections. Currently, quite a lot of the information in the results sections is really methods or introduction sentences. I understand this is difficult based on the format of having the methods section last, but I would prefer these results parts were condensed to just present the important results points here, rather than including too much description and justification of the experimental process involved. I’m guessing this was revised from a combined submission format. Regardless I think that most readers without the specific technical background will find it harder to understand results that are also interspersed with methods, since methods tend to be quite field-specific, and will be more interested in the results rather than the methods used to obtain them.

We apologize for that. We have now paid particular attention to eliminate the unnecessary technical details (now provided in the material and methods section) in order to facilitate reading. We hope the results section is now easier to read and that the train of ideas still flows well.

5. Can the authors comment in the Discussion on how important they predict the MSH4 gene to be in stabilizing meiosis in allopolyploid *B. napus*, and how generalizable these results might be to other allopolyploids? What about the relation of the results to the extra copy of ZIP1 at the Ph1 locus in wheat? I would like to hear more (i.e. expand this Discussion section) about whether any of these

previous results are related in any way to the current findings related to the MSH4 gene – do the genes interact or fulfil similar functions? Can the authors predict what we would expect to find in other polyploid species, or what genes they think are potentially the “major players” in polyploid stabilization?

To be honest, we can only speculate here and we are concerned that the reader could mistake these speculations as over-interpretations. This being said, we have followed reviewer 1’s suggestions and enriched our discussion as requested. Below are provided these and some additional information to answer reviewer1’s questions.

How important can MSH4 gene be in stabilizing meiosis in allopolyploid *B. napus*?

As shown in the manuscript, there are, as expected, two copies of *MSH4* in *B. napus*; these copies are transcribed and functional (in cv. *Tanto* at least). Based on our results, it is very likely that any mutation knocking-out one of the two copies will be selected for to increase fertility, since crossover formation between homoeologous chromosomes is still ongoing (at a low frequency) in *B. napus* (Lloyd et al., 2018; Higgins et al., 2018). Such degenerative mutations, which occur stochastically, may have simply not happened yet in this species that formed only 7500 years ago.

As now stated in the discussion, our results “, which are consistent with pervasive *MSH4* duplicate loss, would indicate that MSH4 is potentially a “major player” in the meiotic adaptation of allopolyploids. However, combined with the results from wheat (in which ZIP4 is the main actor), they could suggest that modulation of the entire the ZMM pathway, or at least different part of it, could contribute to stabilize meiosis in allopolyploids.” This is the hypothesis we are defending and discussing extensively in the manuscript (in order to follow reviewer 1’s recommendation).

How generalizable these results might be to other allopolyploids?

Our phylogenomic survey clearly shows that *MSH4* (and most *ZMMs*) returns to a single copy too systematically to be the result of chance. Compared to our previous study (Lloyd et al., 2014) we have extended our study from 10 to ~90 plant species (plus 95 fungi and 39 animals) and from 14 to more than 40 WGD events in plants (plus 12 additional WGDs across all fungi and animals). Even with this sharp increase, the pattern has remained exactly the same: a single gene encodes MSH4 in all species other than the most recent polyploids. According to some authors, such a consistent pattern of gene loss meets the expectation for deterministic duplicate loss (Edger and Pires 2009). If this is true, then this would suggest that our results are fairly generalizable.

[REDACTED]

What about the relation of the results to the extra copy of ZIP4 at the Ph1 locus in wheat? I would like to hear more (i.e. expand this Discussion section) about whether any of these previous results are related in any way to the current findings related to the MSH4 gene – do the genes interact or fulfil similar functions?

In all species analysed so far, including *Arabidopsis* and rice in plants, MSH4 and ZIP4 work in the same pathway of crossover formation. However, it is very likely that they do not act on the same step within the pathway or on the same recombination intermediates. For example, it is unclear whether MSH4 and ZIP4 *directly* interact with one another, although ZIP4 has been shown to interact directly (through yeast two hybrid) with MSH5, the obligate heterodimeric partner of MSH4, in *Saccharomyces cerevisiae* (de Muyt et al., 2018). The MSH5 / MSH4 heterodimer acts as a meiosis-specific sliding clamp stabilizing single-end invasion intermediates that form during early stages of recombination and subsequently bind to Holliday junctions to facilitate crossover formation. De Muyt et al. (2018) recently proposed that ZIP4 could contribute to “stabilize the Zip2–Spo16 complex” and/or « coordinate the action of the Zip2–Spo16 and Msh4–Msh5 complexes on the branched DNA structure to convert D loops into highly stable recombination intermediates, the SEIs, and then the dHJs”. This would suggest that ZIP4 act upstream of MSH4 in yeast. This hypothesis is consistent with cytological analyses showing that ZIP4 still localizes on chromosomes when MSH4 is missing in this species (Shinohara et al. 2008).

It is difficult to generalize these findings, as conflicting evidence has been obtained in other species. For example, Luo et al. (2013) observed in rice that “OsMSH5 can be loaded onto meiotic chromosomes in *Oszip4*, *Osmer3*, and *hei10*” while “those ZMM proteins cannot be localized normally in the absence of OsMSH5”. Some discrepancies have also been observed within plants. In *Arabidopsis*, for examples, *Atzip4* single mutant and double *Atmsh4Atzip4* double mutant show the same mean number of chiasmata per PMCs in *Arabidopsis*, indicating that they belong to the same epistatic group (Chelysheva et al., 2007). By contrast, the mean chiasma frequency in *Osmsh5 Oszip4* double mutant was very close to that in *Osmsh5* single mutant, but much less than that in the single *Oszip4* mutants of rice (Luo et al., 2013).

To follow reviewer 1’s recommendation, we have now added to the discussion the following paragraph:

“It is not clear whether MSH4 and ZIP4 act on the same step of the ZMM pathway, or even that their specific role is conserved between species. For example, although no ZMM crossovers can form when either MSH4 or ZIP4 are absent in *Arabidopsis*^{8,25}, rice meiosis can admit some residual ZMM-dependent events in *oszip4* mutant⁹. Likewise, MSH4-MSH5 heterodimer is thought to act upstream of ZIP4 in rice^{26,36}, but it seems to be the opposite in yeast³⁷. Finally, we do not know why inter-homoeologue crossover suppression result from *MSH4* duplicate loss in *B. napus* (this study) but *ZIP4* duplication in wheat⁴. These differences call for further investigation into the effect of the loss/gain of individual ZMM duplicates on crossover formation between homologous and homoeologous chromosomes in recent allopolyploids.”

Can the authors predict what we would expect to find in other polyploid species, or what genes they think are potentially the “major players” in polyploid stabilization?

According to our phylogenomic survey of ZMM in plants, MSH5, MER3 and ZIP4 are the proteins that show a pattern of duplicate loss most closely resembling that of MSH4. This could indicate that loss of all but one *MSH5*, *MER3* and *ZIP4* copies is deterministic too. On the other hand, HEI10 is the only ZMM that has been shown to be dosage-sensitive (in *Arabidopsis*; Ziolkowski et al., 2017). It is also the ZMM protein that has retained more duplicates following WGDs. These two attributes can increase the potential variation in HEI10 functional dosage and most importantly lead to more intermediate phenotypic classes that could prevent CO formation between homoeologues while levels of homologous COs are ensured in an optimal way. In the light of the above, it is difficult to predict which ZMMs “are potentially the “major players” in polyploid stabilization”. Therefore, the final answer can only be delivered by practical experiments.

To follow reviewer 1's recommendation, we have now added to the discussion the following paragraph:

“Our results could lead to considering a simple and general route for meiotic adaptations in allopolyploids: i.e. through reduction in the efficiency of the ZMM pathway that would be substantial enough to prevent inter-homoeologue crossover formation but not to compromise the occurrence of (at least) one crossover per homologue (i.e. the obligate crossover). If that is the case, then there would be no need to acquire new meiotic function(s) to sort divergent chromosomes, as the basic machinery would be sufficient to achieve this outcome. In addition, beneficial variants of pre-existing meiotic genes could occur with greater ease, or in greater proportion, as they would be mainly loss-of-function alleles: e.g. dominant-negative mutations that poison ZMM multiprotein complexes (e.g. MSH4-MSH5³⁸⁻³⁹ or MSH5-ZIP4³⁷) or nonsense / missense mutations that suppress or knock down the function of one gene.”

6. In *Arabidopsis*, selective sweeps found several genes potentially responsible for restoration of genome stability via reduction of crossover number in tetraploids relative to diploids. Can the authors elucidate the differences between the genes and mechanisms involved between prevention of non-homologous crossovers and reduction in crossover frequency? Are both likely to operate in allopolyploids, or is the latter predicted to be autopolyploid specific?

This is a good point. We have now added to the discussion the following section: “our result suggest that reduced cellular availability of MSH4 is unlikely to contribute autopolyploid meiosis stabilization, where the challenge is to prevent multivalent formation between chromosomes with little degree of differentiation. In this context, reducing the absolute number of crossover events, rather than improving their stringency (i.e. homologous vs homoeologous), plays a greater role⁴⁰⁻⁴². The invariability of homologous crossover numbers with varying *MSH4* copy numbers (Figures 3, Supplementary Figure 10 and 11) is not conducive to this process. This difference justifies why genome scans in diploid and autotetraploid *Arabidopsis arenosa* populations revealed that differentiation (rather than duplicate loss) of structural components of meiotic chromosomes (rather essential factors for ZMM pathway) likely drives autopolyploid stabilization⁴¹.”

Figures and Tables

7. Figure 2, panel 3 in the top row – does the cluster of chromosomes in the middle of the picture represent bivalents or univalents? Is a clearer image available, or is this a function of the mutant phenotype? The same image is in Supplemental Figure 7

We apologize for this image. Unfortunately, we don't have a clearer metaphase I image for that genotype. We do have some diakinesis that allows the mixture univalent and bivalents to be seen but, for the sake of consistency and because diakinesis is already shown in a supplemental figure (figure S11h), we decided to include this unclear metaphase I image. For this new version, we moved the images of the A¹A¹C²C² genotype to supplemental materials (Figure S8) as proposed in point 17.

8. Are parts A and B of Figure 3 associated with any statistical significance? Why is “Genome” there in part A? If the point is just to demonstrate that there are no differences between categories, I don't think it's necessary to include Parts A and B at all. Just Part C is fine if you want to show the double mutant has a major effect.

Sorry for this confusing labelling. Now we have elaborated a new supplemental figure (figure S7) that includes statistical information on qPCR results and pyrosequencing. For qPCR, we ran an ANOVA with two fixed effects (Genotype and Plate) and the difference in threshold cycle between *MSH4* and *UBC21* as dependent variable. Pairwise comparisons of samples after ANOVA were done using post-

ANOVA Tukey's test. In the euploids, this analysis suggest that the $A^+A^+C^+C^+$ genotype shows a slightly higher level of *MSH4* transcripts than most the other genotypes (including WT *Tanto*). By contrast, no significant differences were observed for the allohaploids. For pyrosequencing, we ran the Marascuillo procedure to compare multiple proportions. In euploids, we observed that the $A^1A^1C^+C^+$ genotype shows a slightly higher contribution of the C copy than all the other genotypes (that are not different from one another). Again, no significant differences were observed for the allohaploids. We have now modified the different charts to provide the assessment of statistical significance for all comparisons (see supplementary Figure 9).

Although we did observe some statistically significant differences, we believe that they do not support any obvious decay or transcriptional compensation between functional and mutant *msh4* alleles. For example, the $A^+A^+C^+C^+$ genotype and WT *Tanto* plants, which both only carry WT *MSH4* alleles, are statistically different while there is no difference between WT *Tanto* and the completely null $A^1A^1C^1C^1$ mutant genotype with regards to *MSH4* transcript accumulation. Likewise, we did not find any significant differences in the allohaploids (neither for the relative contribution of *BnaA.MSH4* vs *BnaC.MSH4* nor for their summed expression) contrary to the euploids. It is thus likely that the rare and small differences we observed for some comparisons are due to random sampling error alone.

Apart from that, we have modified the figure to make clearer the meaning of genomic DNA, which was present in the graphic to show that our pyrosequencing works as the proportion is nearly 50% / 50% as expected.

9. I also don't really think it's so critical to show parts A and B of Figure 4. For Figure 4 have you considered aligning parts C and D over each other, so that the number of crossovers is then matched by an example picture for this mutant type above it? That would make a great single combination figure if it can be done effectively.

Yes, this is a very good idea. We modified the figure 4 accordingly and now it is much more visual and clearer. Thank you for this point.

10. Is it the case that not all of the mutants were screened for the same things? The figures don't seem to match with respect to inclusion of the different mutant types/combinations.

Yes, it is the case, for instance we did not score chiasma frequencies on all the euploid genotypes given that HEI10 immunolocalization is less likely to lead to crossover underestimation.

11. There are no x-axis or y-axis labels on some figures.

Thank you. We have now double-checked all figures and made the necessary corrections.

12. Typo in Figure 1 – “sinlge”

Sorry for this typo. There is no typo now in the new figure 1.

13. Table 1 – based on the number of cells counted, it's statistically invalid to include such a high degree of accuracy as is indicated by the number of decimal places. Even one decimal place is a bit high to be honest, but I'd suggest just going with maximum 2 significant figures.

A fair point. It is now corrected.

14. The title for Table 1 is not descriptive enough for the information to stand on its own – needs species name, basic mutant description etc.

Yes, it was not descriptive enough. The new title is : “Crossover reduction in euploid and allohaploid *B. napus msh4* mutants”.

15. It took me a really long time to find and understand the key for Figure 1, this is really not so clear. I recommend that parts B – D of this figure move to the supplementary information, and that the key format for the indicator of whether the gene copies were duplicated or not is presented using just straight lines instead of curved, and in a table-like format next to the age of WGD events key, preferably in the top right corner. Why are there two > symbols instead of one in the WGD timing key?

We apologize for that. We have now completely modified Figure 1 to make it easier to understand. We now provide bar charts to indicate the number of angiosperms showing one to six complete copies of the genes encoding each ZMM protein and pie charts to show the age of the retained duplicates, as estimated by the age of the WGD they originate from. We still present a phylogeny of MSH4 protein in angiosperms because this is, in our mind, the simplest solution to show (i) that there are some fractionated copies, (ii) that these copies are degenerating (according to the branch length) and (iii) which species display these copies. As proposed by reviewer 1, we have moved the phylogenies of MSH4 in fungi and animals in a Supplemental file (Supplemental Figure 2). Likewise the phylogenies of all other ZMMs are provided in Supplemental Figure 1.

16. Most of the text is talking about results in the allohaploids, so it seems like it would make sense to merge Table 1 with Supplementary Table 3 and include this in the main manuscript.

This is a good idea and so we did in the new Table 1, which is now much more informative.

17. I really like Supplemental Figure 5, although I’d like it better if the wild-type and double mutant meiosis stages were lined up next to each other for ease of comparison, and clearly labelled on the figure as to which was which. Any chance of including this in the main text, maybe by merging with Fig. 2 and cutting one of the two mutant types out from the existing figure 2 to the supplementary? Which double mutant is being presented in Fig S5?

Yes, the cytological comparison is much easier this way. It is done now for the new Figure 2. Thank you for this suggestion. It also makes sense to merge in this same figure, the comparison of cytological stages (part a) with the MLH1/HEI10 immunolocalization (part b). Finally, we decided to keep two separate parts for the sake of accuracy as Tanto and $A^+A^+C^+C^+$ (though they have the same MSH4 genotype) are not exactly the same.

Reviewer #2 (Remarks to the Author):

This is an interesting paper that addresses the role of a conserved recombination factor, MSH4, in meiotic recombination in a polyploid plant (*Brassica napus*). The paper opens with a consideration of the documented phenomenon where meiotic genes are known to reduce to single copy more rapidly than other genes, following whole genome duplication (WGD). As a case in point they present analysis of MSH4, which conforms to this pattern. The authors present a nicely performed phylogenetic analysis on this point - but I think it would be interesting to expand this more generally to other ZMM factors. For example, is the same trend evident for MSH5, HEI10, SHOC1, PTD1, ZIP4 etc?

This is a very good point. We have now provided an assessment of gene copy number for all plant ZMMs (in angiosperms) that confirms our previous finding (Lloyd et al., 2014). As now explained in the results: "Four ZMMs show a conspicuous pattern of rapid duplicate gene loss following WGDs (Figure 1; Supplemental Figure 1) with a single gene encoding MER3, MSH4, MSH5 and ZIP4 proteins in almost all angiosperms other than the most recent polyploids (i.e. that formed <10 000 years ago). In paleopolyploids that have returned to a diploid state post-WGD, the presence of duplicates for these four ZMMs remains an exception usually limited to those with the next most recent WGDs: e.g. MSH5 in *Glycine max* (5-9 million years old WGD 26); MER3 and ZIP4 in *Linum usitatissimum* (5-13 million years old WGD 27) (Figure 1; Supplemental Figure 1). By contrast, a slightly higher number of duplicates is observed among the genes encoding SHOC1, PTD and, most importantly, HEI10 (Supplemental Figure 1). The most striking example of precipitous ZMM duplicate loss remains MSH4, which shows only one intact copy in all diploid species (Figure 1)."

As stated above, this extended analysis is provided in different ways: i.e. using bar and pie sector charts that summarize the number and age of the retained copies, respectively, and using the phylogenies of the proteins that provide an assessment of their relationships and point to the presence of fractionated copies (usually scattered around the phylogenies, as expected because they have accumulated a lot of changes compared to their original homoeologue).

Building from this premise the authors then perform genetic analysis of MSH4 in *Brassica napus* with increasingly fewer functional copies of the gene - ie they compare all combinations of *msh4* loss of function alleles. The assay for meiotic recombination they perform is cytogenetic - specifically they assess the number of univalent, rod and ring form bivalents at metaphase I, in addition they score MLH1 and HEI10 foci, which are markers of Class I interfering crossovers. The authors show that reducing to even a single functional copy of MSH4 is sufficient to show a wild type level of crossover, although it is hard to assess from these data whether the landscape along the chromosomes is the same. They also perform expression analysis to show that there is an absence of compensatory changes in gene expression when MSH4 dosage is reduced. This conclusion is to some extent expected as where tested *msh4* is known to be a recessive mutation in other species.

This is true but, in our mind, it was worth validating with *B.napus msh4* mutants.

The most interesting aspect of the paper involves the authors generating AC haploid plants where the chromosomes lack true homologs to pair and interact with. However, in these plants some pairing and recombination is evident between homeologous chromosomes. In this case the authors do see a much stronger sensitivity to MSH4, which reductions in MSH4 dosage causing much greater reductions in cytological crossover. Can the authors say anything about the likely locations of homeologous crossovers along the chromosomes from their data, and whether this varies with reducing MSH4 activity?

This is a very difficult and delicate task. To answer reviewer2's question, we went back to the cytological images that we took and scored the position of chiasmata based on the shape of the chromosomes. As shown in the figure below, we observed a slight but significant difference between euploids and allohaploids (with chiasmata being more distally located in the latter) but no clear difference among allohaploids showing different combinations of wild type and mutant alleles.

Although we only scored clear and unambiguous open (rod) bivalents, reliable scoring proved to be very difficult, in particular because many bivalents in allohaploids were asymmetrical (as expected as they are formed between homoeologous chromosomes; see below). We are not confident enough with these results to provide them in the manuscript, and are developing further systematic studies to adequately answer this question and integrate with previous results from our lab. For example, we have shown that, in the WT, the rate of crossover formation between homoeologous chromosomes is positively correlated with that of crossovers between homologues (Nicolas et al., 2012; Lloyd et al. 2018). We are currently pursuing new opportunities to use state-of-the-art technologies and improve the resolution of these comparisons, which to date has remained quite low. This is a prerequisite before one can test specifically whether the location of inter-homoeologue crossovers varies with MSH4 dosage.

May be reviewer 2 also wonders about the location of the very rare crossovers that are formed between homoeologous chromosomes in the *msh4* completely defective allohaploid mutants (A^1C^1 and A^1C^2). As discussed in Blary et al. (2018), we believe that these crossovers may have occurred within shared homologous regions carried out by otherwise homoeologous chromosomes. These regions, called "Homoeologous exchanges", result from crossover formation between homoeologous chromosomes. Over time, following chromosome segregation, only one exchanged region becomes fixed within the population, which leads to the replacement of one chromosomal region (which is lost) with a duplicate of the corresponding homoeologous region. Although no "Homoeologous exchanges" has been characterized so far in the cultivar used in that study (cv. Tanto), there is no reason to believe that this variety would be an exception. However, the absence of any HE characterized (in terms of position, size, chromosomes of origin...) in *Tanto* prevents the effective testing of the hypothesis we have formulated.

These findings implicate that ZMM dosage, in this case via MSH4, has a strong effect on homeologous recombination. The authors quite rightly relate this to recent findings that the wheat Ph1 locus contains a duplication of the ZIP4 ZMM gene, and that this locus has a strong effect on homologous

vs homeologous recombination. Therefore, the authors findings have important relevance for regulation of recombination in hybrid situations, which are common in plants. It may also be worth mentioning that HEI10 dosage is also known to regulate levels of homologous recombination in Arabidopsis (Ziolkowski 2017).

Yes, that's right. Ziolkowski et al. 2017 is clearly relevant and now cited in the paper.

Overall the manuscript and data are clearly presented and well written.

Thank you

Minor point: The title and abstract do not mention the species that this study focuses on (B.napus) - please add this information!

Good point! Done!

REVIEWERS' COMMENTS:

Reviewer #1 (Remarks to the Author):

I am very happy with the changes the authors have made and the revised manuscript; congratulations to the authors on a very nice study.

I have only the minor suggestion that the first two paragraphs of the Discussion are now a little bit too much like a reiteration of the Introduction, and should be condensed a bit so as to focus instead more on the implications of the results rather than repeating the justifications for the study.

Reviewer #2 (Remarks to the Author):

Thank you - the reviewers have addressed my concerns fully.

REVIEWERS' COMMENTS:

Reviewer #1 (Remarks to the Author):

I am very happy with the changes the authors have made and the revised manuscript; congratulations to the authors on a very nice study.

Thank you!

I have only the minor suggestion that the first two paragraphs of the Discussion are now a little bit too much like a reiteration of the Introduction, and should be condensed a bit so as to focus instead more on the implications of the results rather than repeating the justifications for the study.

This is true. We have now completely removed the first paragraph of the discussion that starts without reiterating the justifications for the study. We have also slightly modified the second paragraph to focus on the implications of the results rather than the way we got them.

Reviewer #2 (Remarks to the Author):

Thank you - the reviewers have addressed my concerns fully.

Thank you!